# PROMPT-GUIDED VISUAL PERCEPTION FOR EFFICIENT TRAINING-FREE VIDEO LLMS

## ABSTRACT

Vision-language large models have achieved remarkable success in various multi-modal tasks, yet applying them to video understanding remains challenging due to the inherent complexity and computational demands of video data. While training-based video-LLMs deliver high performance, they often require substantial resources for training and inference. Conversely, training-free approaches offer a more efficient alternative by adapting pre-trained image-LLMs models for video tasks without additional training, but they face inference efficiency bottlenecks due to the large number of visual tokens generated from video frames. In this work, we present a novel prompt-guided visual perception framework (abbreviated as *Free Video-LLM*) for efficient inference of training-free video LLMs. The proposed framework decouples spatial-temporal dimension and performs temporal frame sampling and spatial RoI cropping respectively based on task-specific prompts. Our method effectively reduces the number of visual tokens while maintaining high performance across multiple video question-answering benchmarks. Extensive experiments demonstrate that our approach achieves competitive results with significantly fewer tokens, offering an optimal trade-off between accuracy and computational efficiency compared to state-of-the-art video LLMs.

## 1 INTRODUCTION

Recently, vision-language models (VLMs) have rapidly revolutionized multi-modal understanding and generation, enabling models to comprehend and produce responses from both visual and textual inputs (visual inputs include images, videos, etc). Due to the vast amount of image-text data and foundational models like CLIP and large language models (LLMs) such as GPT Brown (2020) and LLaMA Touvron et al. (2023), image-LLMs Zhang et al. (2024a); Liu et al. (2023); Achiam et al. (2023); Chen et al. (2024) have already made significant progress. InstructBLIP Dai et al. (2023) extends the capabilities on new visual tasks by incorporating instruction-aware features. LLaVA Liu et al. (2023) generate multimodal language-image instruction-following data using GPT-4 Achiam et al. (2023), demonstrating impressive multimodel chat abilities. The evolution of VLMs has also given rise to video-based LLMs that are specifically tailored for video understanding. These methods, as discussed by Tang et al. Tang et al. (2023), can be categorized into two primary approaches based on their training strategies: training-based video LLMs and training-free video LLMs.

In the training-based video LLMs, AV-LLM Shu et al. (2023) and Vid2Seq Yang et al. (2023) trained all the parameters in the LLM, which can be resource-intensive. Other main stream methods in this fine-tuning category either externally(*e.g.*, Q-former) or internally fine-tune the bridge between the video encoder and the LLM, or adopt a phased fine-tuning approach for connective adapters and insertive adapters. Video-LLMs, such as Video-ChatGPT Maaz et al. (2024), Chat-UniVi Jin et al. (2024), and MovieChat Song et al. (2024a), primarily focus on general question-answering (QA) or captioning tasks. They utilize video encoders to produce video embeddings, which they then decode into text outputs based on given prompts or instructions. Video-LLaMA Zhang et al. (2023) integrates a video Q-former to learn video-language correspondence and an audio Q-Former to learn audio embeddings for the LLM. Additionally, Chat-UniVi Jin et al. (2024) employs a set of dynamic visual tokens to uniformly represent both images and videos.

Despite the remarkable success of training-based video LLMs, training-free video understanding models have emerged as an attractive alternative due to the high cost of training large-scale video models. Training-free methods leverage pre-trained image-based models and minimally adapt them for video tasks without additional training on video data. Models such as IG-VLM Kim et al. (2024) and FreeVA Wu (2024) have demonstrated the potential of transforming video frames into a format compatible with high-performance VLMs. These approaches utilize parameter-free temporal aggregation techniques and composite image grids, enabling robust zero-shot performance on video question-answering tasks. These methods offer promising results without requiring time-consuming and resource-intensive video-specific training.

However, existing training-free video LLMs, struggle with efficiency issues due to the large number of visual tokens generated by video frames. Video data consists of multiple frames, significantly increasing the number of visual tokens that need to be processed by the model. As the input sequence length increases, the computational cost also scales, especially in transformer architectures where self-attention and feed-forward networks dominate resource usage. This results in slow inference speeds and high memory consumption, limiting the practical application of video LLMs in real-time or resource-constrained environments. Reducing token count without compromising performance is a key challenge in scaling video LLMs efficiently.

To address these challenges, we propose a prompt-guided visual perception framework (abbreviated as *Free Video-LLM*) that significantly improves the efficiency of training-free video LLMs. Our method introduces prompt-guided temporal and spatial sampling, which reduces the number of visual tokens based on the specific requirements of the input prompt. By discarding redundant frames and focusing only on the regions and time segments relevant to the prompt, our model achieves superior performance across multiple video QA benchmarks with a significantly reduced computational burden. Table 1 compares several prominent open-source models, our proposed method uniquely possesses training-free capabilities, inference efficiency, and effective video understanding, demonstrating a comprehensive advantage in the field. The proposed Free Video-LLM approach not only enhances the model's efficiency but also maintains competitive accuracy compared to existing state-of-the-art video LLMs, offering a balanced solution for scalable video understanding tasks.

Table 1: Comparison of the existing representative open-sourced vision-language models.

| Method | Training-free | Inference-efficient | Video understanding |
|---|:---:|:---:|:---:|
| Image-LLM (*e.g.,* LLaVA) | ✓ | ✗ | ✗ |
| Training-based Video LLM (*e.g.,* Video-LLaVA) | ✗ | ✗ | ✓ |
| Training-free Video LLM (*e.g.,* IG-VLM) | ✓ | ✗ | ✓ |
| Ours | ✓ | ✓ | ✓ |

## 2 RELATED WORKS

In this section, we briefly revisit the related works including image-LLMs, video-LLMs (especially training-free video LLMs).

### 2.1 IMAGE-LLMS

The rapid advancement of image-language models (image-LLMs) can be attributed to two key factors: the foundational work of CLIP Radford et al. (2021), which introduced a shared representation space for vision and language, demonstrating strong zero-shot capabilities and robust performance across various computer vision benchmarks; and the emergence of powerful Large Language Models (LLMs) like GPT Brown (2020) and LLaMA Touvron et al. (2023), which can be further enhanced through instruction tuning Peng et al. (2023). Flamingo Alayrac et al. (2022) excels in few-shot learning by seamlessly integrating pre-trained vision and language models and high-quality interleaved multimodal data. BLIP-2 Li et al. (2023a) utilizes a lightweight Querying Transformer to connect modalities.

Further progress on image-LLMs has been achieved through multimodal instruction tuning. LLaVA Liu et al. (2023) uses GPT-4 Achiam et al. (2023) to generate robust multi-modal instruction data, pre-training on image-text pairs and fine-tuning for end-to-end multimodal understanding.

InstructBLIP Dai et al. (2023) extends the capabilities of BLIP-2 with instruction-aware features, while mPLUG-Owl Ye et al. (2023) employs a two-stage modular approach to enhance task performance across different modalities. MiniGPT-4 Zhu et al. (2023) aligns a frozen visual encoder with a frozen LLM (Vicunna Peng et al. (2023)) using one projection layer and showcasing enhanced capabilities such as detailed image descriptions and creative storytelling. MiniGPT-5 Zheng et al. (2023) extends MiniGPT-4 to output text interleaved with images.

## 2.2 Video LLMs

Building on the foundation of LLM and image-LLMs, video-language models (video-LLMs) ia also rapidly developed. FrozenBiLM Yang et al. (2022) leverages frozen bidirectional language models for zero-shot video question answering, achieving leading performance on zero-shot VideoQA without the need for manual annotations. VideoChat Li et al. (2023b) and Video-LLaMA Zhang et al. (2023) both utilize dual streams to handle audio and visual signals. Specifically, Video-LLaMA integrates Q-Former for these two streams, whereas VideoChat incorporates a video embedder alongside a perception tools for captions, and introduce a video-centric multimodal instruction fine-tuning dataset. Video-ChatGPT Maaz et al. (2024) utilizes a pretrained visual encoder to extract both spatial and temporal features from videos by averaging frame-level features. These features are then projected into the input space of large language models (LLMs). Additionally, it also contributes a high-quality dataset of 100,000 video-instruction question-answer pairs. Chat-UniVi Jin et al. (2024) use a set of dynamic visual tokens to uniformly represent images and videos tokens. PLLaVA Xu et al. (2024a) introduce a post-training weight fusion methods to alleviate forgetting phenomenon during multi-modality fine-tuning.

## 2.3 Training-free Video LLMs

In contrast to these tuning methodologies for Video LLMs, there is a growing interest in exploring training-free video LLMs, specifically how existing image model architectures can be minimally adapted to accommodate video inputs. IG-VLM Kim et al. (2024), as a pioneer in this exploration, transforms videos into single composite image grids, enabling the direct application of a high-performance VLM without the need for video-data training. FreeVA Wu (2024) also investigates the potential of leveraging offline image-LLMs as training-free video assistants by simply adding a parameter-free temporal aggregation, achieving strong performance in zero-shot video question-answering tasks Chen & Dolan (2011); Caba Heilbron et al. (2015); Xu et al. (2016), even surpassing video instruct-tuning models. Meanwhile, SLOWFAST-LLAVA Xu et al. (2024b) utilizes a dual-stream approach to efficiently capture spatial and temporal video features, demonstrating competitive performance on various video benchmarks without requiring additional training.

# 3 Approach

## 3.1 Preliminaries

The image-LLM has a remarkable progress in the past two years due to the large amount of image-text pairs data and image instruction data. The representative open-sourced image-LLMs like LLaVA Liu et al. (2023) and InternVL Chen et al. (2024) obtain high performance on various image conversation, description and reasoning tasks. The image-LLMs take one image $I$ as input, and the visual encoder $g_V$ (including the projector) extracts the image features and converts into language embedding tokens:

$$H_I = g(I), \quad (1)$$

where $H_V \in \mathbb{R}^{N \times D}$, $N$ is the number of visual tokens per frame, and $D$ is the token embedding dimension. Then the image tokens $H_I$ and the text tokens $H_T$ are fed into the LLM $f$ for generating responses:

$$Y = f([H_I, H_T]). \quad (2)$$

## 3.2 Computational Burden of Video LLM

The training-free video LLM can leverage the well-trained image-LLMs for video without training on any data. Given a video, a number of frames are usually uniformly extracted and forms a sequence

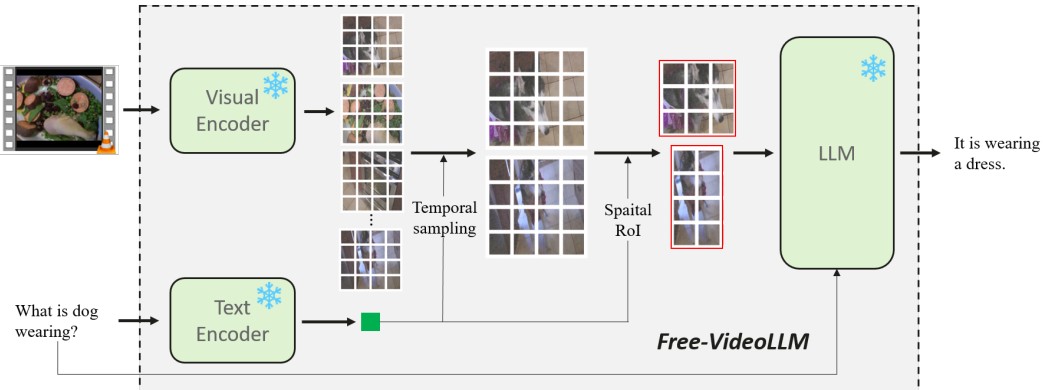

Figure 1: Illustration of the proposed *Free Video-LLM* with prompt-guided visual perception.

of images $\{I_1, I_2, \cdots, I_T\}$ where $T$ is the number of frames. These images are fed into the visual encoder to obtain visual tokens:

$$X_V = g_V(\{I_1, I_2, \cdots, I_T\}), \tag{3}$$

$X_V \in \mathbb{R}^{T \times N \times D}$. Then the training-free video LLMs directly use LLMs to receive visual tokens $X_V$ and text prompt to generate responses for video understanding:

$$Y = f([H_V, H_T]). \tag{4}$$

This simple pipeline is intuitive yet effective Wu (2024); Kim et al. (2024).

However, the video usually consists of many frames, that is, the number of visual tokens will be significantly large, proportional to the product of the frame number and the image size $L \propto TN$. The input sequence length directly influences the computational cost and inference efficiency of LLM. In transformer architecture, the computational cost is mostly occupied by the self-attention and feed-forward network. The self-attention module performs token-to-token relation computation and requires a $O(L^2)$ computational cost. The feed-froward network makes nonlinear transformation of each token with the computation budget proportional to the sequence length. Compared to image-LLMs, the computational complexity of video-LLMs will increase by an order of magnitude, which will significantly reduce the inference speed and efficiency of video-LLMs, affecting the actual usage experience.

### 3.3 PROMPT-GUIDED VISUAL PERCEPTION

We introduce the prompt-guided visual perception for efficient video LLMs, by temporally and spatially sampling visual information respectively, as demonstrated in Figure 1.

**Prompt-guided Temporal Sampling**  The video is usually captured in a high frame rate, resulting in large redundancy in the neighbor frames. In order to reduce the number of frames and maintain the discriminative information, we propose to temporally sample the prompt-related frames. The previous video-LLMs take all the visual tokens generated by the visual encoder as inputs for LLMs. However, this approach does not consider the text context for different questions or tasks, which may introduce useless tokens during inference. For instance, given a video depicting two women shopping in a supermarket, one might ask how many apples they bought, while another person could ask whether they bought eggs. Different questions necessitate the model to focus on different periods of the video. On the other hand, different regions of the video are also required for different questions. For example, one may inquire about the number of persons, while another may ask about the color of the left woman's T-shirt. Here, we introduce prompt-guided spatial and temporal sampling to reduce the number of visual tokens for different input prompts.

In the temporal dimension, we use prompt features to guide the selection of temporal frames most relevant to the prompt. In particular, We utilize the text encoder $g_T$ that is matched with the visual encoder (such as the two in CLIP) to extract prompt features:

$$F_P = g_T(P), \tag{5}$$

where one prompt sequence $P$ will be represented as a vector $F_P \in \mathbb{R}^D$. The visual tokens of video frames after the visual encoder contains the information inside each frame, and the tokens are in the same subspace of the prompt features. The visual features after global average pooling $F_V \in \mathbb{R}^{T \times D}$ are used to represent each frame. We calculate the relation score (*i.e.,* cosine similarity) between the frame features and the prompt feature:

$$s_i = \frac{F_{Vi}}{\|F_{Vi}\|_2} \cdot \frac{F_P}{\|F_P\|_2}, \tag{6}$$

where $i = 1, 2, \cdots, T$ is the frame temporal index. We sample the frames based on the similarities so as to maintain the most related frames to the prompt and discard the useless ones.

**Prompt-guided Spatial Sampling**    As mentioned above, different questions require the model to focus on a specific part of the spatial regions for answering. We propose to utilize the prompt information to guide the selection of regions of interest (RoI) in the video. For the visual tokens of a frame, we reshape them back to the spatial size as $X_{Si} \in \mathbb{R}^{H \times W \times D}$ where $H \times W$ is the feature map size. For a feature vector in spatial position $(h, w)$, its relation score with the prompt feature is calculated:

$$s_{h,w} = \frac{X_{Si,h,w}}{\|X_{Si,h,w}\|_2} \cdot \frac{F_P}{\|F_P\|_2}, \tag{7}$$

where $1 \leq h \leq H$ and $1 \leq w \leq W$. Suppose we need to box out a RoI with an area that is $\alpha \times$ the original area where $0 < \alpha < 1$. The top-$K$ tokens with the largest similarity scores are selected, and their positions are $\{(h_1, w_1), (h_2, w_2), \cdots, (h_K, w_K)\}$ where $K = \alpha HW$. The center coordinates of the RoI can be obtained by the mean of these positions:

$$h_c = \frac{1}{K} \sum_{k=1}^{K} h_k, \tag{8}$$

$$w_c = \frac{1}{K} \sum_{k=1}^{K} w_k. \tag{9}$$

The height and width of the RoI is calculated by

$$H' = \sqrt{\alpha} H, \tag{10}$$

$$W' = \sqrt{\alpha} W. \tag{11}$$

With the center coordinates and box size, we can easily crop the RoI from the frame feature maps, and reshape them as a sequence of tokens as the video representation for the specific prompt. The obtained compact visual tokens are concatenated with the text prompt embeddings as the inputs of LLMs. In this way, the training-free video LLMs can process a video with fewer tokens and enjoy more efficient inference.

## 4 EXPERIMENTS

### 4.1 BENCHMARKS AND IMPLEMENTATION DETAILS

**Benchmarks**    As our method is training-free, we directly evaluate the proposed methods on open-ended video understanding and question-answering benchmarks, including MSVD-QA Chen & Dolan (2011), MSRVTT-QA Xu et al. (2016), ActivityNet-QA Caba Heilbron et al. (2015) and TGIF-QA Jang et al. (2017). The GPT APIs are utilized to assess the model accuracy and response quality. Following the previous works Wu (2024); Xu et al. (2024a;b), GPT-3.5-Turbo-0125 version is used for fair comparison.

**Base Models**    The image MLLM we utilized as base models is LLaVA-v1.6 Liu et al. (2023; 2024). The two versions with different model sizes are used, *e.g.,* 7B and 34B. Both the visual encoder and text encoder are from OpenAI's CLIP-L-14. The pretrained weights of LLaVA-v1.6[1] and CLIP-L[2] can be downloaded on HuggingFace. We fix all the pretrained weights of the base models and only slightly modify the hyperparameters.

---

[1] https://huggingface.co/collections/liuhaotian/llava-16-65b9e40155f60fd046a5ccf2
[2] https://huggingface.co/openai/clip-vit-large-patch14-336

**Implementation Details**   The input video is resized to 336×336 for matching the CLIP-L visual encoder. Each frame will be transformed into 24×24 tokens. The proposed method will squeeze the frame number and crop an RoI for efficient video understanding. The scaling factor of RoPE (rotary position embedding) in LLaVA-v1.6 is set as 2 to enable processing context with 8192 tokens. All the models are implemented using PyTorch.

## 4.2 MAIN RESULTS

In Table 2, we evaluate several models with a focus on efficiency, as reflected by the number of visual tokens (second column), which directly impacts the computational cost and memory requirements. Our method demonstrates notable efficiency, using only 1026 and 2600 visual tokens across two configurations, which is significantly fewer than competing models such as IG-VLM (3456 tokens) and SF-LLaVA (3680 tokens). Despite this reduced token count, our model consistently achieves competitive or superior performance across various QA benchmarks. For instance, in the MSVD-QA task, our method scores 76.8/4.0 with 1026 tokens, outperforming FreeVA (73.8/4.1 with 2304 tokens). Overall, the average accuracy and score of our method are comparable to other models with fewer inference tokens.

Table 2: Main results of the proposed method and comparison with other training-free video LLMs. All models are 7B-level and with CLIP-L visual encoder. The two numbers in each cell are 'Accuracy/Score' respectively. The **bold** numbers indicate that our method requires significantly fewer tokens during inference.

| Model | #visual tokens | Size | MSVD -QA | MSRVTT -QA | ANet -QA | TGIF -QA | Avg |
|---|---|---|---|---|---|---|---|
| LLaVA-NeXT-Image Zhang et al. (2024b) | 2304 | 7B | - | - | 53.8/3.2 | - | - |
| FreeVA Wu (2024) | 2304 | 7B | 73.8/4.1 | 60.0/3.5 | 51.2/3.5 | - | - |
| Free Video-LLM (ours) | **513** | 7B | 74.9/3.9 | 60.8/3.4 | 51.2/3.4 | 73.8/3.9 | 65.2/3.7 |
| Free Video-LLM (ours) | **1026** | 7B | 76.8/4.0 | 62.9/3.5 | 53.9/3.4 | 75.6/4.0 | 67.3/3.7 |
| IG-VLM Kim et al. (2024) | 3456 | 7B | 78.8/4.1 | 63.7/3.5 | 54.3/3.4 | 73.0/4.0 | 67.5/3.8 |
| SF-LLaVA Xu et al. (2024b) | 3680 | 7B | 78.1/4.0 | 64.1/3.4 | 55.3/3.4 | 78.4/4.1 | 69.0/3.7 |
| Free Video-LLM (ours) | **2648** | 7B | 78.2/4.0 | 65.6/3.6 | 54.8/3.4 | 77.8/4.1 | 69.1/3.8 |

This efficiency does not come at the expense of performance; rather, it highlights the ability of our approach to achieve optimal trade-offs between token usage and task accuracy. On the TGIF-QA benchmark, our model with 2600 tokens achieves a competitive score of 78.5/4.1, slightly surpassing IG-VLM (73.0/4.0 with 3456 tokens) and SF-LLaVA (77.3/4.0). Table 3 shows the results on larger LLM, *i.e.,* 34B. The similar efficiency gain to that of 7B models can be seen. Overall, our model provides a balanced approach to both computational efficiency and task performance, offering substantial improvements in resource allocation without sacrificing accuracy.

Table 3: Main results of the proposed method and comparison with other training-free video LLMs. All models are 34B-level and with CLIP-L visual encoder. The **bold** numbers indicate that our method requires significantly fewer tokens during inference.

| Model | #visual tokens | Size | MSVD -QA | MSRVTT -QA | ANet -QA | TGIF -QA | Avg |
|---|---|---|---|---|---|---|---|
| IG-VLM Kim et al. (2024) | 3456 | 34B | 79.6/4.1 | 62.4/3.5 | 58.4/3.5 | 79.1/4.2 | 69.9/3.8 |
| SF-LLaVA Xu et al. (2024b) | 3680 | 34B | 79.3/4.1 | 67.0/3.6 | 58.8/3.5 | 80.2/4.2 | 71.3/3.9 |
| Free Video-LLM (ours) | **2648** | 34B | 79.2/4.1 | 67.2/3.7 | 59.0/3.5 | 79.8/4.2 | 71.3/3.9 |

## 4.3 COMPARISON WITH SOTA METHODS

In Table 4, we present a comparison between our model and state-of-the-art (SOTA) methods, with a particular emphasis on the trade-off between accuracy and efficiency, as reflected by the number of visual tokens and performance on four video question-answering (QA) benchmarks. Our method

utilizes 2600 visual tokens, which is more efficient than IG-VLM (3456 tokens) and SF-LLaVA (3680 tokens), while achieving highly competitive results across all evaluated tasks.

On the MSVD-QA benchmark, our model scores 78.2/4.0, being comparable with SF-LLaVA (78.1/4.0), but with much fewer tokens, demonstrating a substantial gain in efficiency. A similar trend is observed on the MSRVTT-QA benchmark, where our model achieves 65.6/3.6, outperforming SF-LLaVA's 64.1/3.4 while using fewer visual tokens. Additionally, our model shows strong performance on the ANet-QA and TGIF-QA benchmarks, with faster inference and lower computational cost. This balance of high accuracy and reduced computational load highlights the strength of our approach in delivering comparable or superior performance with significantly fewer resources.

In comparison to other training-free models, such as FreeVA (2304 tokens) and IG-VLM (3456 tokens), our method offers a superior balance, outperforming FreeVA on all tasks and matching IG-VLM's performance with fewer tokens. These results demonstrate that our approach achieves an optimal trade-off between efficiency and performance, making it an excellent choice for scalable video understanding tasks.

Table 4: Comparison with SOTA video LLMs on the four benchmarks. The **bold** numbers indicate that our method requires significantly fewer tokens during inference.

| Model | #visual tokens | Size | Visual Encoder | MSVD -QA | MSRVTT -QA | ANet -QA | TGIF -QA |
|---|---|---|---|---|---|---|---|
| Trained models | | | | | | | |
| Video-LLaMA Zhang et al. (2023) | - | 7B | CLIP-G | 51.6/2.5 | 29.6/1.8 | 12.4/1.1 | - |
| Video-LLaVA Lin et al. (2023) | - | 7B | ViT-L | 70.7/3.9 | 59.2/3.5 | 45.3/3.3 | 70.0/4.0 |
| Vista-LLaMA Ma et al. (2023) | - | 7B | CLIP-G | 65.3/3.6 | 60.5/3.3 | 48.3/3.3 | - |
| VideoChat Li et al. (2023b) | - | 7B | CLIP-G | 56.3/2.8 | 45.0/2.5 | 26.5/2.2 | 34.4/2.3 |
| VideoChat2 Li et al. (2024) | - | 7B | UMT-L | 70.0/3.9 | 54.1/3.3 | 49.1/3.3 | - |
| MovieChat Song et al. (2024a) | - | 7B | CLIP-G | 75.2/3.8 | 52.7/2.6 | 45.7/3.4 | - |
| Video-ChatGPT Maaz et al. (2024) | - | 7B | CLIP-L | 64.9/3.3 | 49.3/2.8 | 35.2/2.7 | 51.4/3.0 |
| Video-LLaMA2 Cheng et al. (2024) | - | 7B | CLIP-L | 70.9/3.8 | - | 50.2/3.3 | - |
| PLLaVA Xu et al. (2024a) | - | 7B | CLIP-L | 76.6/4.1 | 62.0/3.5 | 56.3/3.5 | 77.5/4.1 |
| Training-free models | | | | | | | |
| FreeVA Wu (2024) | 2304 | 7B | CLIP-L | 73.8/4.1 | 60.0/3.5 | 51.2/3.5 | - |
| IG-VLM Kim et al. (2024) | 3456 | 7B | CLIP-L | 78.8/4.1 | 63.7/3.5 | 54.3/3.4 | 73.0/4.0 |
| SF-LLaVA Xu et al. (2024b) | 3680 | 7B | CLIP-L | 78.1/4.0 | 64.1/3.4 | 55.3/3.4 | 78.4/4.1 |
| Free Video-LLM (ours) | **2648** | 7B | CLIP-L | 78.2/4.0 | 65.6/3.6 | 54.8/3.4 | 77.8/4.1 |

## 4.4 Ablation Studies

**Inference Speed** We compare the inference speed of three representative training-free video LLMs in Table 5. Two important metrics for measuring the inference speed of large models are pre-filling latency and output speed. Pre-filling latency represents the time taken by the model to generate the first output token after receiving the input. A lower pre-filling latency means the model can begin producing results faster. Output speed refers to the rate at which the model generates subsequent tokens after the first one is produced. A higher TPS indicates that the model can output tokens more quickly once the generation process has started. The inference speed is evaluated on an NVIDIA V100 GPU with standard transformers framework. Our Free Video-LLM outperforms the others with fewer visual tokens (2,648), the fastest pre-filling latency (0.578 seconds), and the highest output speed (20.4 TPS). In comparison, IG-VLM and SF-LLaVA have higher visual token counts, slower pre-filling latency, and lower output speeds.

**Prompt-guided Temporal Sampling** We evaluate the effectiveness of the proposed prompt-guided temporal sampling on MSVD task. The base image-LLM is LLaVA-v1.6-7B. The methods evaluated include uniform temporal sampling baseline, prompt-guided temporal sampling, and a combination of uniform and prompt-guided temporal sampling. Uniform temporal sampling with 3 frames and 864 visual tokens achieves an accuracy of 71.7 on MSVD. This method provides a baseline for comparison. Prompt-guided temporal sampling also uses 3 frames and 864 visual tokens

Table 5: The inference speed of the compared training-free video LLMs. Pre-filling latency means time to the first output token. TPS denotes tokens per second.

| Method | #visual tokens | Avg Accuracy | Pre-filling latency | Output speed |
|---|---|---|---|---|
| IG-VLM Kim et al. (2024) | 3456 | 73.0/4.0 | 0.894 s | 18.2 TPS |
| SF-LLaVA Xu et al. (2024b) | 3680 | 77.3/4.0 | 0.961 s | 17.9 TPS |
| Free Video-LLM (ours) | **2648** | 77.8/4.1 | 0.578 s | 20.4 TPS |

but yields a higher score of 75.0. This indicates that incorporating prompt information to guide the temporal sampling process can lead to improved performance. Finally, the combination of uniform and prompt-guided temporal sampling, with 6 frames and 1728 visual tokens, achieves a score of 77.2. This suggests that a hybrid approach may further enhance results.

Table 6: The effectiveness of prompt-guided temporal sampling.

| Method | #frames | #visual tokens | MSVD-QA |
|---|---|---|---|
| Uniform temporal sampling | 3 | 864 | 71.7 |
| Prompt-guided temporal sampling | 3 | 864 | 75.0 |
| Uniform + Prompt-guided temporal sampling | 6 | 1728 | 77.2 |

**Prompt-guided Spatial RoI Cropping**  We conduct experiments to evaluate the proposed prompt-guided spatial sampling for RoI cropping. The experiments involve uniform temporal sampling and prompt-guided temporal sampling, with and without the addition of RoI. Uniform temporal sampling and prompt-guided temporal sampling serves as baselines without RoI cropping, with 3 frames and 864 visual tokens resulting in MSVD accuracies of 71.7 and 75.0 respectively. When adding RoI to prompt-guided temporal sampling, with an RoI ratio of 0.6 and 3 frames, the number of visual tokens is reduced to 513 and the MSVD accuracy is 74.9. Replacing the proposed RoI method with adaptive average pooling will decrease the accuracy to 73.8 with the same number of visual tokens. These experiments demonstrate combining prompt-guided sampling with RoI can enhance performance, showing the effectiveness of prompt-guided spatial RoI cropping.

Table 7: The effectiveness of prompt-guided spatial RoI cropping.

| Method | RoI ratio | #frames | #visual tokens | MSVD-QA |
|---|---|---|---|---|
| Uniform temporal sampling | - | 3 | 864 | 71.7 |
| Prompt-guided temporal sampling | - | 3 | 864 | 75.0 |
| Prompt-guided temporal sampling + AvgPool | 0.6 | 3 | 513 | 73.8 |
| Prompt-guided temporal sampling + RoI | 0.6 | 3 | 513 | 74.9 |

Table 8 presents the results of our experiments on the effect of RoI ratio during prompt-guided spatial cropping on MSVD-QA performance. The findings reveal a general trend of increasing accuracy with higher RoI ratios. Specifically, an RoI ratio of 0.4 yields 360 visual tokens and an MSVD-QA score of 74.1, while increasing the RoI ratio to 0.5 results in a modest improvement to 74.2 with 408 tokens. As the RoI ratio increases beyond 0.6, performance

Table 8: The effect of RoI ratio. The setting is prompt-guided temporal sampling + RoI.

| RoI ratio | #frames | #visual tokens | MSVD-QA |
|---|---|---|---|
| 0.4 | 3 | 360 | 74.1 |
| 0.5 | 3 | 408 | 74.2 |
| 0.6 | 3 | 513 | 74.9 |
| 0.7 | 3 | 600 | 74.8 |
| 1.0 | 3 | 864 | 75.0 |

stabilizes, with the scores hovering around 74.8 to 74.9, despite an increase in the number of visual tokens. This suggests that while a larger RoI allows for more comprehensive spatial information, there may be diminishing returns in accuracy beyond a certain threshold (*i.e.,* 0.6). Overall, these results indicate that optimizing the RoI ratio is crucial for enhancing model performance in video question-answering tasks, highlighting the effectiveness of our prompt-guided approach in balancing token utilization and accuracy.

## 4.5 VIDEO QA EXAMPLES

We also show the video QA examples to analyze the advantages and disadvantages of the method more intuitively. Figure 2 summarizes example object detection results from the MSVD-QA benchmark, comparing the outputs of SF-LLaVA and our proposed method. For the first video, both models correctly identify that someone is folding a square green paper, with SF-LLaVA scoring slightly higher (5 vs. 4). In the second video, both models describe multiple men diving into a swimming pool, with neither able to accurately determine the number of divers, resulting in the same lower score of 2. Overall, while both models perform similarly on these tasks, our method is more efficient with much fewer tokens.

| Video | SF-LLaVA results | Our results |
|---|---|---|
|  | R: a person is seen folding a piece of square green paper. The hands of the person are visible in the key frames, with the fingers and thumb in the process of folding the paper. The paper is green and square-shaped, and the person appears to be carefully folding it. The context of the video is not provided, but it seems to be a simple demonstration or instructional video on how to fold a piece of paper. | R: there are two people interacting with a piece of square green paper. One person is holding the paper and appears to be folding it. The exact action of folding the paper is not clear from the image provided. |
| Q: who folded a piece of square green paper? GT: someone | ✓, score: 5 | ✓, score: 4 |
|  | R: there are multiple men diving into a swimming pool. The exact number of men diving cannot be determined without more information. | R: there are multiple instances of men diving into a swimming pool. The exact number of men diving cannot be determined without more specific information about the video. |
| Q: how many men dive into a swimming pool? GT: five | ✗, score: 2 | ✗, score: 2 |

Figure 2: Example video question answering results on MSVD-QA benchmark.

## 5 CONCLUSION

In conclusion, this paper introduces a novel framework for efficient inference of training-free video LLMs, named Free Video-LLM. The proposed method addresses the computational challenges associated with video understanding by introducing a prompt-guided visual perception approach that significantly reduces the number of visual tokens processed by the model. Through temporal and spatial token sampling techniques tailored to the input prompt, Free Video-LLM achieves a remarkable reduction in computational burden without compromising accuracy. Extensive experiments demonstrate that our method not only matches but often exceeds the performance of current state-of-the-art video LLMs, while using a fraction of the visual tokens. This represents a substantial step forward in the practical application of video LLMs, offering a competitive balance between accuracy and computational efficiency. The prompt-guided framework paves the way for more scalable and efficient video understanding systems, potentially transforming real-time and resource-constrained environments.

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
