# OpenReview forum: "Prompt-guided Visual Perception for Efficient Training-free Video LLM"
_ICLR.cc/2025/Conference — ICLR 2025 Conference Withdrawn Submission_

### Official Review · Reviewer_ovuG · 2024-10-21

**Soundness:** 2
**Presentation:** 2
**Contribution:** 1
**Rating:** 3
**Confidence:** 4

**Summary:**

This paper introduces an efficient inference framework for training-free video LLMs. The presented prompt-guided visual perception framework reduces the number of visual tokens in two ways: First, it samples the prompt-related frames to temporarily drop unnecessary frames. Second, it selects a RoI to make the video LLMs focus on a smaller spatial area. This method, called Free Video-LLM, achieves competitive results while using fewer visual tokens. The authors show the effectiveness of Free Video-LLM on multiple QA benchmarks using the LLaVa-v1.6 model.

**Strengths:**

- The proposed prompt-guided visual perception framework is straightforward and intuitive. The two prompt-guided sampling techniques are all based on cosine similarity, which is easy to understand.
- The paper reads well and is easy to follow. It looks well-organized, with the required tables and equations placed throughout.
- The proposed method shows a good trade-off between accuracy and efficiency regarding the number of visual tokens.

**Weaknesses:**

- [W1] Although the authors argue that the proposed method can focus only on the regions and time segments relevant to the prompt (l.74), they have not provided any empirical results supporting this argument. Does the prompt-guided temporal sampling technique really sample only the prompt-related frames? Does the prompt-guided spatial sampling select the most informative area? The presented results seem only anecdotal, and I believe the authors should provide direct evidence and deeper analysis.
- [W2] The prompt-guided spatial sampling technique is too naive. It finds a center position by averaging the top-K tokens and builds a RoI around the computed center position with a fixed ratio of 0.6. This has three problems: First, it might have a high degree of center bias, meaning that the center position of the RoI is highly biased to the center of the frame. This is because the proposed technique simply takes the mean value of the top-K tokens. Second, the fixed RoI ratio may not be optimal for other tasks or datasets. More advanced techniques can have an adaptive RoI ratio depending on the context of each video. Third, a single RoI might not be enough for some cases. For example, some tasks might require multiple regions that lie scattered. The proposed technique assumes that the prompt-related region is a single, connected area, which may not be valid for many other cases. I believe that more advanced techniques should be developed for better spatial sampling.
- [W3] The authors have used only one architecture, LLaVa-v1.6. The proposed sampling techniques may not work for other MLLMs. For example, it seems like they work only with CLIP models, which have aligned visual and text encoders.

**Questions:**

Could you also discuss the limitations of the proposed framework/techniques?

---

### Official Review · Reviewer_EnA5 · 2024-11-01

**Soundness:** 3
**Presentation:** 3
**Contribution:** 3
**Rating:** 6
**Confidence:** 4

**Summary:**

The paper proposes prompt-guided temporal and spatial sampling for visual tokens. By virtue of being able to focus only on query-relevant regions, the method achieves QA performances efficiently.

**Strengths:**

The core idea of the paper is to avoid unnecessary temporal frames and spatial areas as per the prompt, which leads to compute efficiency and a training-free video-LLM framework. We see that the framework produces competitive or sometimes better results on QA tasks, that too without requiring any training.

**Weaknesses:**

For the problem statement of a training-free video LLM, the proposed framework achieves what it aims for.
A bigger question for a video-llm field, in general, would be how to assess a model if it does temporal reasoning. The datasets shown in this paper aren't the best uses cases. I get that this is an a focus of this particular paper.

**Questions:**

Is it possible to use recent egocentric datasets and benchmarks for assessing QA performance of this Free Video LLM method?

---

### Official Review · Reviewer_KtJ5 · 2024-11-03

**Soundness:** 2
**Presentation:** 3
**Contribution:** 2
**Rating:** 5
**Confidence:** 4

**Summary:**

This paper proposed a training-free inference method for video-LLM, namely Free Video-LLM, aiming to enhance the model efficiency without sacrificing the model performance. Specifically, it reduced the number of visual tokens to the LLM by selecting important frames in time and important tokens (with a ROI pooling) in space per frame with top-K prompt-guided selection, where the prompt was from the CLIP text encoder. Experiments were conducted with LLaVA-v1.6 and OpenAI CLIP-L-14 on 4 video QA benchmarks, showing performance on par with other baseline video-LLM models.

**Strengths:**

1. The paper has a clear statement and is well organized.
2. Figure 1 is well illustrated, and the proposed method is intuitive to understand.
3. Experiments show that Free Video-LLM performs on par with baseline models with fewer visual tokens, and this improves the output speed of the LLM.

**Weaknesses:**

**Major**:

1. The novelty of this paper is somewhat limited since using sparse video tokens is not new in the literature, e.g. [a] selects top-K visual tokens, [b] adopts token merging, [c] compresses visual tokens into context and content tokens using text decoder.  It is unclear how the performance and computation of the proposed token reduction method compared to these baselines under the same number of visual tokens.

2. (Sec 3.3) The proposed Prompt-guided Spatial Sampling crops an ROI box (with alpha ratio of the image size) with a center coordinate obtained by the mean of top-K visual patch tokens. However, why not using these top-K visual tokens directly but using the ROI box?  It is also unclear how the proposed method is compared to using the image center as the center coordinate (with the same alpha). Note that all the ablations should be conducted with the same Video-LLM for fair comparison.

3. The paper addresses the model efficiency as the number of input tokens into the LLM. It is true that longer sequence requires more computations in the LLM part. However, there is no computation analysis on the token selection part (before the LLM).

4. (Table 6) The comparisons between prompt-guided temporal sampling and uniform temporal sampling only considers a low-frame-rate scenario, i.e. 3 frames. It is unclear whether the prompt-guided method is still important when more frames are taken (e.g. Table 2).

**Minor**:

5. The proposed method only keeps visual tokens that are relevant to the query question, which is helpful for the speed in this run, but if the user wants to ask follow-up questions, the visual tokens need to be processed again.

6. Table 6 & 7 are not referenced in paragraphs.

7. When citing prior works, it would be more readable to put authors’ names in bracket, e.g. Chat-UniVi Jin et al. (2024) -> Chat-UniVi (Jin et al., 2024).

8. Typo: H_V in L154 -> H_I

**Ref**:
- [a] Li et al., SViTT: Temporal Learning of Sparse Video-Text Transformers, CVPR 2023.

- [b] Bolya et al., Token Merging: Your ViT but Faster, ICLR 2023.

- [c] Li et al., LLaMA-VID: An Image is Worth 2 Tokens in Large Language Models, ECCV 2024.

**Questions:**

In addition to the questions in the weakness section, can the authors also provide a density map for the center coordinates obtained with Eq. (8-9)?

---

### Official Review · Reviewer_fhbU · 2024-11-04

**Soundness:** 2
**Presentation:** 2
**Contribution:** 2
**Rating:** 5
**Confidence:** 5

**Summary:**

Free Video-LLM introduces a training-free, prompt-guided visual perception framework that improves inference efficiency in video LLMs by decoupling spatial-temporal dimensions through temporal frame sampling and spatial RoI cropping based on task-specific prompts, reducing visual tokens while maintaining high performance across video question-answering benchmarks.

**Strengths:**

The paper introduces Free Video-LLM, a training-free framework that enhances efficiency in video LLMs through prompt-guided temporal and spatial sampling, reducing visual tokens without compromising performance, and achieving superior results across video QA benchmarks with lower computational costs.

**Weaknesses:**

Inconsistency in Reported Results: The paper mentions that "On the TGIF-QA benchmark, our model with 2600 tokens achieves a competitive score of 78.5/4.1, slightly surpassing IG-VLM (73.0/4.0 with 3456 tokens) and SF-LLaVA (77.3/4.0)." However, 2600 tokens and the corresponding accuracy values are not found or do not match those presented in the tables. This inconsistency makes it difficult to verify the claimed performance improvements. Clarification is needed. Could you clarify the origin of the reported score of 78.5/4.1 for the 2600-token model on the TGIF-QA benchmark? This score is mentioned in the text but does not seem to match any data in the tables. Could you ensure consistency between the text and the tables?

Lack of Direct Comparisons and Generalizability: In Tables 2 and 3, each model utilizes a different baseline Vision-Language Model (VLM), which complicates the assessment of the efficiency gains from temporal sampling and spatial ROI cropping. To accurately evaluate the effectiveness of the proposed token reduction methods, it is essential to compare the performance of the original LLAVA 1.6 model to that of LLAVA 1.6 enhanced with the proposed techniques. Additionally, applying the proposed token reduction method to various baseline VLMs such as LLaVA-NeXT would demonstrate its generalizability and effectiveness across different models. Without these direct comparisons, it is challenging to understand the true impact and applicability of the proposed approach. Could you include an experiment or table comparing the original LLAVA 1.6 model with your enhanced LLAVA 1.6 model, detailing both performance metrics and token counts? This would help in assessing the efficiency gains of your proposed methods. Could you also demonstrate the effectiveness of your token reduction method on another Vision-Language Model baseline, such as LLaVA-NeXT, to support the claims of generalizability?

Insufficient Discussion of Related Work: The concept of question/text-guided token reduction has been previously introduced in works like LVNet [1] and VideoAgent [2]. The paper does not adequately compare the similarities and differences between the proposed method and these existing approaches. A thorough discussion of how the proposed method builds upon or differs from prior work is necessary to establish its novelty and contribution to the field. Without this comparison, the paper's unique contributions remain unclear. Could you add a subsection in the Related Work or Discussion section that explicitly compares your method with LVNet and VideoAgent? This would be helpful to highlight key similarities and differences, as well as clarify how your approach advances beyond these existing methods.

References:
[1] Park, Jongwoo, et al. "Too Many Frames, Not All Useful: Efficient Strategies for Long-Form Video QA." arXiv preprint arXiv:2406.09396 (2024).

[2] Wang, Xiaohan, et al. "Videoagent: Long-form video understanding with large language model as agent." arXiv preprint arXiv:2403.10517 (2024).

**Questions:**

I left questions in the weakness section

---

### Official Review · Reviewer_kmoh · 2024-11-04

**Soundness:** 3
**Presentation:** 3
**Contribution:** 2
**Rating:** 3
**Confidence:** 4

**Summary:**

The submission presents a way to convert a pre-trained image-language model into a "video"-language model. The submission presents a method to pack image-tokens of a video into a stream of visual tokens that can be fed into a pre-trained VLM. At it's core the method filters input frames using a CLIP embedding, and uses spatial nearest neighbors to define a prompt-specific region of interest. The results show some improvement over baselines.

**Strengths:**

- The paper is mostly well written and easy to follow.
- The technical contribution is well explained, concise, and clearly separated from prior approaches.
- The method is explained in enough detail, and likely easy to reproduce

**Weaknesses:**

- **Framing:** The comparison, and motivation through prior work is incomplete. Throughout the submission, image-language models are presented as single-image models. For example L.150 "The image-LLMs take *one* image I as input". This is clearly no longer the case. Yes the original LLaVA architecture supported just one image, but LLaVA-next [Zhang et al. (2024b)], InternVL (cited as supporting evidence for L.150), EMU2, Mantis, etc all support multiple images at least at inference time. This significantly changes the framing of the paper and will require a major rewrite. The current framing is: "Convert a video into a image-size token stream, and feed this through the VLM". A more accurate framing would be "Use an existing multi-image VLM and subsample the visual tokens, right? This puts a much larger burden on the runtime evaluation.
- **Evaluation:** The evaluation seems incomplete.
  - One major claimed advantage of the proposed method is faster runtime. However, runtime is never explicitly compared. Instead, the submission presents `# tokens` as a proxy metric. This is not an ideal proxy, as it completely ignores the additional cost of running CLIP, and potential batching or pipeline inefficiencies. I would like to see a complete breakdown of runtimes and comparisons to prior work on the rebuttal.
  - Another issue are baselines. I really appreciate the ablations in Tables 6, 7 and 8, however they are not quite apples-to-apples, as the number of visual tokens varies drastically (for Table 8 this is understandable as it is part of the experiment, but Table 6 and 7 should control for it). Both Tables 6 and 7 would be much more informative as a graph with varying number of tokens for each baseline. This would also provide the reader with a calibrated baseline if the llava-1.6 performance on the input videos (uniform sampling of *all* input tokens).
  - Finally, it is quite standard for VLMs to encode the image once, and answer multiple questions at inference time. How would the proposed framework perform under this more standard setting? Presumably, caching the image tokens will no longer work, as they depend on the prompt. Ideally, the rebuttal shows some evidence that caching vision tokens does not fully take away the advantage of the proposed method.
- **Technical:** The proposed CLIP-based sampling technique might lead to highly correlated inputs. Since CLIP is an embedding-based model, it might always find the k-closest images to the text-prompt. It seems likely that these k-closest frames are all the same (if the video is sampled at a high enough frequency). In fact, this may be happening, the results in Table 6 show that combining uniform and CLIP-based sampling works significantly better.

**Questions:**

- What would the story/pitch look like if multi-image VLMs were taken into consideration?
- What is the complete breakdown of runtimes and comparisons to prior work?
- Can you produce a graph-based version of Tables 6 and 7, `#tokens` vs performance?
- How do the performance characteristics (in concrete numbers) change if multiple questions are asked about the same video (allowing for kv-caching)?
- How much correlation is there in the sampled frames? How do the authors prevent near duplicate frames from being sampled?

---

### Note · Authors · 2024-11-13

I have read and agree with the venue's withdrawal policy on behalf of myself and my co-authors.